# REACH THE REMOTE NEIGHBORS: DUAL-ENCODING TRANSFORMER FOR GRAPHS

## ABSTRACT

Despite recent successes in natural language processing and computer vision, Transformer suffers from the scalability problem when dealing with graphs. Computing node-to-node attentions is infeasible on complicated graphs, e.g., knowledge graphs. One solution is to consider only the near neighbors, which, however, will lose the key merit of Transformer that attends to the elements at any distance. In this paper, we propose a new Transformer architecture, named dual-encoding Transformer (DET), which has a structural encoder to aggregate information from near neighbors and a semantic encoder to focus on useful semantically close neighbors. The two encoders can be incorporated to boost each other's performance. Our experiments demonstrate that DET achieves superior performance compared to the respective state-of-the-art attention-based methods in dealing with molecules, networks and knowledge graphs.

## 1 INTRODUCTION

Transformer has become one of the most prevalent neural models for natural language processing (NLP) (Vaswani et al., 2017; Devlin et al., 2019). The self-attention mechanism leveraged by Transformer has already been extended to graph neural networks (GNNs), e.g., GAT (Velickovic et al., 2018) and its variants (Wu et al., 2019; Vashishth et al., 2020; Chen et al., 2021b; Kim & Oh, 2021). Nevertheless, these models only consider the near (usually one-hop) neighbors, which may violate the original intention of Transformer that attends to the elements at distant positions.

Recently, Graphormer (Ying et al., 2021) starts to leverage the standard Transformer architecture for graph representation learning and has achieved superior performance on many benchmarks. However, in its scenarios of graph property prediction, the datasets are small graphs (e.g., small molecules). The full node-to-node attention leveraged by Graphormer makes it inapplicable to large graphs with millions of nodes, such as knowledge graphs (KGs) or social networks. The same problem also appears in the computer vision (CV) area, yet has recently been tackled by patching pixels to patches and then to windows in a hierarchical fashion (Dosovitskiy et al., 2021; Liu et al., 2021b). These works inspire us to explore the possibility of using one universal Transformer architecture as the general backbone to model graphs of different sizes.

In addition to many self-attention-based methods considering only one-hop neighbors (Schlichtkrull et al., 2018; Wu et al., 2019; Ye et al., 2019; Chen et al., 2021b; Kim & Oh, 2021), some existing works introduce multi-hop (usually 2- or 3-hop) neighbors (Abu-El-Haija et al., 2019; Sun et al., 2020). However, they still concentrate on the local information and fail to obtain useful information from the remote nodes. Capturing the remote correlations is one of the most important characteristics of Transformer, because the rich context not only boosts the performance but also avoids over-fitting for local information. For example, attending to the distant nodes may be helpful even on a highly homophilic graph, considering the existence of enormous missing links (Ciotti et al., 2016).

In this paper, we propose a dual-encoding Transformer (DET). In DET, we consider two types of neighbors, i.e., structural neighbors and semantic neighbors. Structural neighbors are the near neighbors leveraged by most existing GNNs (Vashishth et al., 2020; Chen et al., 2021b; Kim & Oh, 2021). By contrast, Semantic neighbors, i.e., the remote neighbors, are the semantically close nodes but may be remote from the node of interest in structure.

Figure 1: Overview of DET. Structural neighbors are local neighbors connected with the node of interest on the graph, while semantic neighbors are remote nodes with similar semantics to the node of interest. The two encoders focus on encoding different aspects of neighboring information, and thus are capable of complementing each other.

Figure 1 shows the basic idea of DET. For structural encoding, we use the standard self-attention layer to encode the structural neighbors. For semantic encoding, we use a modified linear attention layer to encode the semantic neighbors. The dual encoding ensures both local aggregation and global connection, and also enables them to benefit from each other through back propagation.

The idea of reaching remote neighbors is inspired by MSA Transformer (Rao et al., 2021) and AlphaFold 2 (Jumper et al., 2021). They query the genetic database to fetch the similar sequences (i.e., proteins) as "family members". The difference is that the family members in DET (i.e., semantic neighbors) are obtained by self-supervised learning rather than asking for external resources. Briefly, we convert this problem to a learning task to find the distant nodes that are as important as local neighbors. We then view local neighbors as positive examples and randomly sampled distant nodes as negative examples, constructing a standard contrastive learning. Furthermore, we propose to use the semantic operator ⊖ to estimate the score between the node of interest and others. It is a learnable function to compel the encoder to value the semantically close nodes.

The proposed DET is capable of achieving superior performance on various graph learning tasks. (1) For graph property prediction, DET outperforms the best-performing methods on the PCQM4M-LSCv1 (Nakata & Shimazaki, 2017) and ZINC (Dwivedi et al., 2020) datasets; (2) For node classification, DET obtains competitive or better performance compared with the state-of-the-art attention-based methods, on several prevalent benchmarks (e.g., Cora, PPI, and ogbn-arxiv) (Yang et al., 2016; Nakata & Shimazaki, 2017; Zitnik & Leskovec, 2017); (3) For KG completion (a.k.a., entity prediction), DET achieves the state-of-the-art performance on both FB15K-237 (Toutanova & Chen, 2015) and WN18RR (Dettmers et al., 2018).

## 2 RELATED WORKS

We split the related literature into three parts: non-local GNNs, self-attention, and position embedding.

**Non-local GNNs** Some methods also investigate how to capture the relationships among disconnected nodes (Pei et al., 2020; Yao et al., 2020; Liu et al., 2021a; Min et al., 2022). Specifically, Geom-GCN (Pei et al., 2020) learns the aggregation purely based on embedding distance, while Non-local-GNNs (Liu et al., 2021a) uses the attention scores from a virtual node to other nodes as a sorting metric to find non-local neighbors. However, they only focus on modeling networks and are evaluated on multi-classification tasks with a few classes. They also do not distinguish between remote nodes and direct neighbors. (Yao et al., 2020; Min et al., 2022) leverage hand-crafted features to find the useful remote nodes, which are less relevant to our work.

**Self-attention** Self-attention-based neural models, such as Transformer, have recently become the *de facto* choice in NLP, ranging from language modeling and machine translation (Devlin et al., 2019; Vaswani et al., 2017) to question answering (Yang et al., 2019; Yavuz et al., 2022) and sentiment analysis (Cheng et al., 2021; Xu et al., 2019a). Transformer has significant advantages over conventional sequential models like recurrent neural networks (RNNs) (Williams & Zipser, 1989; Hochreiter & Schmidhuber, 1997) in both scalability and efficiency.

**Position Embedding**    Position embedding is one of the most important modules of Transformer. Transformer variants in different fields typically customize different designs in this module. For example, ViT and its followers (Dosovitskiy et al., 2021; Fan et al., 2021; Han et al., 2021) sequentially index the patches and encode the indices as 1D position embeddings. SwinTransformer (Liu et al., 2021b;c) proposes the 2D-aware relative position biases, which employs a learnable matrix to record pairwise patch position information in the window.

In addition to the position information, other prior knowledge can also be injected as attention biases or position embeddings into Transformer, which becomes the key to applying Transformer on graphs (Ahn et al., 2021; Chen et al., 2021a;b; Dwivedi & Bresson, 2021; Kreuzer et al., 2021; Ying et al., 2021). For example, GT (Dwivedi & Bresson, 2021) replaces the sinusoidal positional embeddings by the Laplacian eigenvectors. Graphformer (Ying et al., 2021) encodes centrality and shortest path distance into embeddings, and then incorporates them as position embeddings into Transformer. HittER (Chen et al., 2021b) adds the edge type (i.e., relation) information of KGs when encoding entity embeddings.

## 3    METHODOLOGY

In this section, we present the details of DET. We start from the preliminaries and then introduce the dual-encoding process. Finally, we illustrate how to train a DET model.

### 3.1    PRELIMINARIES

We first introduce the terminologies and notations that will be used in the following sections.

**Graph**    We define a graph as $\mathcal{G} = (\mathcal{V}, \mathcal{E})$, where $\mathcal{V} = \{v_1, v_2, ..., v_n\}$ is the node set, and $\mathcal{E} = \{e_1, e_2, ..., e_m\}$ is the edge set. $n$ and $m$ denote the number of nodes and edges, respectively. In practice, different tasks often have more complicated graph structures. For example, molecular graphs and KGs have edge types (i.e., chemical bonds and relations). We do not discuss the details and follow the general setting to process these specific features (Chen et al., 2021b; Ying et al., 2021).

**GNN and Self-attention**    Without loss of generality, we define a GNN as a neural network that learns a group of weights to aggregate the embeddings of the one-hop or multi-hop neighbors for the node of interest. In this sense, self-attention can be naturally treated as a GNN model. Let $\boldsymbol{Q} \in \mathbb{R}^{n \times h}$, $\boldsymbol{K} \in \mathbb{R}^{n \times h}$, $\boldsymbol{V} \in \mathbb{R}^{n \times h}$ denote the query, key, and value matrices, respectively. In this paper, they are the same node embedding matrix. $h$ denotes the hidden layer size. Self-attention calculates the attention scores as follows:

$$\boldsymbol{A} = Softmax(\frac{\boldsymbol{Q}\boldsymbol{K}^{\top}}{\sqrt{h}}), \tag{1}$$

where $\boldsymbol{A} \in \mathbb{R}^{n \times n}$ records the node-to-node attention scores. We then aggregate the node embeddings with the following equation:

$$\boldsymbol{H} = \boldsymbol{A}\boldsymbol{V}, \tag{2}$$

where $\boldsymbol{H} \in \mathbb{R}^{n \times h}$ is the output embedding matrix, with each row denoting the embedding of a node.

**Linear Attention**    The computational complexity of the above dot-product implementation is $\Omega(n^2)$ (without considering the hidden layer size). As the number of nodes increases, the cost becomes unacceptable. GAT (Velickovic et al., 2018) proposes a linear self-attention implementation to mitigate this problem by only considering the one-hop neighbors:

$$\boldsymbol{B}_{ij} = \sigma\big(\mathbf{b}^{\top}(\boldsymbol{W}\mathbf{v}_i \,\|\, \boldsymbol{W}\mathbf{v}_j)\big), \tag{3}$$

where $\boldsymbol{B}_{ij}$ denotes the attention score from the node of interest $v_i$ to a neighbor $v_j$. $\mathbf{v}_i, \mathbf{v}_j \in \mathbb{R}^h$ are the embeddings of $v_i$ and $v_j$, respectively. $\mathbf{b} \in \mathbb{R}^h$ and $\boldsymbol{W} \in \mathbb{R}^{h \times h}$ are weight vector and matrix, respectively. $\sigma$ is the activation function and $\|$ denotes the concatenation operation. The linear attention does not consider the correlations within neighbors, and thus its computational complexity is cut down from $\Omega(n^2)$ to $\Omega(m)$. Note that, it is not necessary to use one-hop neighbors as keys in linear attention. Some existing works also consider multi-hop neighbors (Sun et al., 2020).

Table 1: The occurrence frequency of entities in FB15K-237 and WN18RR, in term of hops.

| Dataset | 1-hop | 2-hop | 3-hop | 5-hop |
|---------|-------|-------|-------|-------|
| WN18RR | 2.7 | 8.9 | 30.5 | 483.8 |
| FB15K-237 | 20.3 | 1781.4 | 64,774.9 | - |

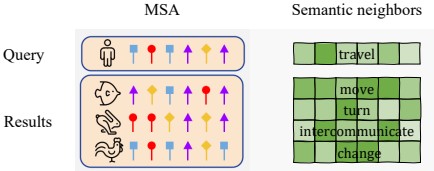

Figure 2: A comparison between MSA and semantic neighbors. The left figure is sliced from Jumper et al. (2021). The right figure is an example from WordNet (Miller, 1995).

---

**Algorithm 1** Dual-encoding Transformer

1: **Input:** graph $\mathcal{G} = (\mathcal{V}, \mathcal{E})$, the main prediction loss $\mathcal{L}_{\mathrm{main}}$, the structural encoder $\mathcal{M}_{\mathrm{st}}$, and semantic encoder $\mathcal{M}_{\mathrm{se}}$;
2: Initialize all parameters;
3: **repeat**
4:     Update the semantic neighbors;
5:     **for each** batch data $(\boldsymbol{X}_{st}, \boldsymbol{X}_{se}, \boldsymbol{Y})$ **do**
6:         $\boldsymbol{H}_{\mathrm{st}} \leftarrow \mathcal{M}_{st}(\boldsymbol{X}_{st})$ (Equation (4));
7:         $\boldsymbol{H}_{\mathrm{se}} \leftarrow \mathcal{M}_{se}(\boldsymbol{X}_{se})$ (Equation (6));
8:         $\boldsymbol{H} \leftarrow H_{\mathrm{st}} \oplus H_{\mathrm{se}}$;
9:         Compute $\mathcal{L}_{\mathrm{sn}}$ (Equation (7));
10:        $\mathcal{L} \leftarrow \mathcal{L}_{\mathrm{main}}(\boldsymbol{H}, \boldsymbol{Y}) + \mathcal{L}_{\mathrm{sn}}$;
11:        Update the parameters according to $\mathcal{L}$;
12:     **end for**
13: **until** the performance on the validation set converges;

---

## 3.2 STRUCTURAL ENCODING

The standard dot-product attention can be easily extended on the small graphs. We add a virtual node $v_c$ (Devlin et al., 2019) as the context node connected with all nodes in $\mathcal{G}$. Then, the output representation for $v_c$ can be regarded as an embedding of $\mathcal{G}$. For the large graphs like KGs or networks, we perform self-attention on the local subgraph $\mathcal{G}_i$ given the node of interest $v_i$. Therefore, the output embedding for node $v_i$ is

$$\mathbf{h}_i^{\mathrm{st}} = \sum_{v_j \in \{v_i\} \cup \mathcal{N}(v_i)} \boldsymbol{A}_{cj} \mathbf{v}_j, \qquad (4)$$

where $\mathbf{h}_i^{\mathrm{st}}$ denotes the output of the structural encoder for $v_i$. $\boldsymbol{A}_{cj}$ denotes the attention score from the context node $c$ to the neighbor $v_j$. $\mathcal{N}(v_i)$ denotes the set of local neighbors (one-hop neighbors in our implementation) for $v_i$. Follow the existing works (Chen et al., 2021b; Ying et al., 2021), we accordingly add the centrality, relation type, or shortest distance path information as special position embeddings to the encoder. The details can be found in Appendix A.

## 3.3 SEMANTIC ENCODING

The structural features sometimes are unreliable for identifying a neighbor. For example, if the node of interest has a large amount of neighbors, many of them inevitably have similar or identical structural features (e.g., shortest path distance to $v_i$). This problem is even more serious when they are all one- or two-hop neighbors. However, if we directly aggregate more-hop nodes, the sheer quantity of available information will overwhelm the neural network.

Table 1 summarizes the average frequency of entities appearing as others' neighbors in different hops. We can find that the two- or more-hop neighbors of a node are shared by many others, which is why current GNNs rarely consider multi-hop neighbors. This phenomenon also reveals the over-smoothing problem to some extent. To make the node of interest more distinguishable to the classifier, weighting its one-hop neighbors is usually reasonable due to the less redundancy. Therefore, we make the following hypothesis:

**Hypothesis 1** *The one-hop neighbors are the most informative features to identify and represent the node of interest.*

Recent successes (Rao et al., 2021; Jumper et al., 2021) in biological science demonstrate that using the information provided by the family members can help protein structure prediction. Specifically, they leverage multiple sequence alignment (MSA) (the results of biological sequence alignment) to make use of the information within an evolutionary family. The protein sequences in a family are assumed to have a common ancestor or an evolutionary relationship. Therefore, they may also share some important sub-structures in protein folding.

If we view the embedding of a node as a kind of sequence, then the remote neighbors used in semantic encoding should have similar embeddings to the node of interest. We illustrate this idea in Figure 2, our basic idea is to view node embeddings as protein amino acid sequences with a fixed length. Then, we can make use of the insight from the well-known MSA Transformer and AlphaFold 2. To this end, we need to find the family members (that share similar evolutional characteristics and relationships) of our "protein". In their original implementation, this step is done by querying an external gene database. In our case, we propose the semantic encoder to find such family members. Therefore, we make the second hypothesis:

**Hypothesis 2** *The distant nodes that have a high embedding similarity with the node of interest are important features for this node.*

In this paper, we estimate the similarity score by a learnable neural function $f_s : \mathbb{R}^h \times \mathbb{R}^h \to \mathbb{R}$:

$$
\begin{aligned}
f_s(\mathbf{v}_i, \mathbf{v}_j) &= \mathbf{v}_i \ominus \mathbf{v}_j \\
&= 1 - \sigma\big((\mathbf{v}_i - \mathbf{v}_j)\boldsymbol{W}_s + b_s\big),
\end{aligned} \tag{5}
$$

where $\ominus$ is the semantic difference operator. In fact, the choice of $\ominus$ is flexible as long as it can reflect the similarity between $\mathbf{v}_i$ and $\mathbf{v}_j$. We use the weighted difference as $\ominus$, which can be easily extended to matrix operation. $\boldsymbol{W}_s \in \mathbb{R}^{h \times 1}$ and $b_s \in \mathbb{R}$ are the weight and bias, respectively. We use Sigmoid as activation to normalize the difference to $(0, 1)$ and then convert it to a similarity score. Therefore, the output embedding for semantic encoding is written as follows:

$$
\begin{aligned}
\mathbf{h}_i^{\text{se}} &= \sum_{v_j \in \mathcal{N}^{\text{se}}(v_i)} \boldsymbol{B}_{ij}\mathbf{v}_j \\
&= \sum_{v_j \in \mathcal{N}^{\text{se}}(v_i)} f_s(\mathbf{v}_i, \mathbf{v}_j)\mathbf{v}_j,
\end{aligned} \tag{6}
$$

where $\boldsymbol{B} \in \mathbb{R}^{n \times n}$ records the node-to-node semantic attention scores estimated by $f_s$ in Equation (5). $\mathcal{N}^{\text{se}}(v_i)$ is the semantic neighbor set. In our implementation, it is sampled from the top candidates (top $10\%$ in our setting) during training.

We then define the semantic neighbor fetching loss to learn $f_s$:

$$
\mathcal{L}_{\text{sn}}(v_i) = -\frac{1}{|\mathcal{N}(v_i)|} \sum_{v_j \in \mathcal{N}(v_i)} \ln\big(f_s(\mathbf{v}_i, \mathbf{v}_j)\big) + \frac{1}{|\mathcal{N}^-(v_i)|} \sum_{v_k \in \mathcal{N}^-(v_i)} \ln\big(f_s(\mathbf{v}_i, \mathbf{v}_k)\big), \tag{7}
$$

where $\mathcal{N}(v_i)$ is the positive example set that includes the local neighbors ( Hypothesis 1) of $v_i$, and $\mathcal{N}^-(v_i)$ is the negative example set in which the negative examples are randomly-sampled distant nodes (Hypothesis 2) of $v_i$. Therefore, the learning process of $f_s$ is self-supervised and can be jointly optimized with the main task loss.

### 3.4 DUAL-ENCODING TRANSFORMER

**Algorithm** We illustrate the implementation of DET in Algorithm 1 and summarize the overall training process as follows: We first initialize the input embeddings and all parameters of DET. For each epoch (or every few epochs), we first draw semantic neighbors from the top $10\%$ candidates (according to Equation (5) for each node. In each batch, we feed the structural encoder with the structural neighbors of the input nodes, and the semantic encoder with the semantic neighbors. We combine the output embeddings of two encoders by weighted addition and jointly minimize the main task loss and semantic neighbor fetching loss.

**Computational Cost** We find the total training time does not increase too much compared with the baselines. The design of the semantic operator $\ominus$ in the semantic encoder is simpler than the linear attention, which only increases a small number of parameters. Although fetching the semantic neighbors needs to iterate all nodes (yields a time complexity of $n^2$), we do not compute them on the fly. Instead, we update semantic neighbors of each node every few epochs, improving both efficiency and robustness. Hence, the overall training time remains at the same level (see Appendix B for the detailed statistics).

Table 2: Graph property prediction results on the PCQM4M-LSCv1 dataset.

| Model | #param. | train MAE | validate MAE |
|---|---|---|---|
| GCN | 2.0M | 0.1318 | 0.1691 |
| DeeperGCN | 25.5M | 0.1059 | 0.1398 |
| GraphSage | - | - | - |
| GIN | 3.8M | 0.1203 | 0.1537 |
| GT | 83.2M | 0.0955 | 0.1408 |
| Graphormer | 47.1M | 0.0582 | 0.1234 |
| DET | 47.1M | **0.0546** | **0.1212** |

Table 3: Graph property prediction results on the ZINC dataset.

| Model | #param. | test MAE |
|---|---|---|
| GCN | 505,079 | 0.367 |
| GraphSage | 505,341 | 0.398 |
| GIN | 509,549 | 0.526 |
| GatedGCN-LSPE | - | 0.090 |
| GAT | 531,345 | 0.384 |
| GT | 588,929 | 0.226 |
| Graphormer | 489,321 | 0.122 |
| DET | 489,562 | **0.113** |

## 4 EXPERIMENT

We conducted experiments on a variety of benchmarks to verify the effectiveness of DET. We uploaded the source code and reported the dataset statistics and parameter settings in Appendix C.

### 4.1 GRAPH PROPERTY PREDICTION

**Datasets** We evaluated DET on the graph property prediction benchmarks PCQM4M-LSCv1 (Hu et al., 2021) and ZINC (Dwivedi et al., 2020). The former is used in the recent Open Graph Benchmark Large-Scale Challenge, while the latter is a popular dataset used to evaluate molecular graph representation learning methods. Considering that the number of nodes in each graph is very small (usually less than 50), we directly perform attention operations on the whole graph. Therefore, we removed the semantic neighbor fetching loss in this experiment.

**Baselines** We compared DET with the state-of-the-art methods: the attention-based GAT (Velick-ovic et al., 2018), GT (Dwivedi & Bresson, 2021) and Graphormer (Ying et al., 2021); and other recently developed GCN (Kipf & Welling, 2017), GraphSage (Hamilton et al., 2017), GIN (Xu et al., 2019b), DeeperGCN (Gilmer et al., 2017), and GatedGCN-LSPE (Dwivedi et al., 2022).

**Results** Table 2 and Table 3 summarize the experimental results measured by mean average error (MAE) on the two datasets. Due to the inaccessibility of the testing data on PCQM4M-LSCv1, we alternatively report the MAE results on the training and validation sets.

Overall, DET outperformed all the baseline methods on PCQM4M-LSCv1. Compared with Graphormer that only considers encoding structural information with Transformer, DET significantly improved the performance, with 6.2% and 7.4% MAE decline on PCQM4M-LSCv1 and ZINC, respectively. Furthermore, the number of model parameters still maintained the same level to that of baselines. We also observed that DET had more significant advantages over other Transformer-based methods on ZINC. Although GatedGCN-LSPE had a better result, we argue that it is feasible to use it as our structural encoder to obtain better performance.

### 4.2 NODE CLASSIFICATION

**Datasets** We evaluated DET on five benchmarks that are generally used for node representation learning. Specifically, Cora, CiteSeer, and PubMed (Yang et al., 2016) are three citation network datasets commonly used in the transductive setting, while PPI (Zitnik & Leskovec, 2017) is a well-used protein-protein interaction dataset in the inductive setting. ogbn-arxiv (Nakata & Shimazaki, 2017) is a large citation network dataset. Most experiment settings follow (Kim & Oh, 2021), and we repeated experiments 100 times on Cora, CiteSeer, and PubMed with random seeds, and 30 times on PPI and ogbn-arxiv, to produce reliable results and ensure a fair comparison.

**Baselines** We selected the attention-base methods GAT (Velickovic et al., 2018), CGAT (Wang et al., 2019a), Graph-Bert (Zhang et al., 2020) and SuperGAT$_{SD}$ (Kim & Oh, 2021) as baseline methods. In addition, other GNN-based methods like GCN (Kipf & Welling, 2017), GraphSage (Hamilton et al., 2017) and GCN+NS (Zheng et al., 2020) were also added for comparison.

Table 4: Node classification results on five benchmarks (accuracy for Cora, CiteSeer, PubMed, and ogbn-arxiv; F1-score for PPI). The results of Graph-Bert are from (Zhang et al., 2020).

| Model | Cora | CiteSeer | PubMed | PPI | ogbn-arxiv |
|---|---|---|---|---|---|
| GCN | 81.5 | 70.3 | 79.0 | 61.5±0.4 | 33.3±1.2 |
| GraphSage | 82.1±0.6 | 71.9±0.9 | 78.0±0.7 | 59.0±1.2 | 54.6±0.3 |
| GCN+NS | 83.7±1.4 | **74.1**±1.4 | - | - | - |
| GAT | 83.0±0.7 | 72.5±0.7 | 79.0±0.4 | 72.2±0.6 | 54.1±0.5 |
| CGAT | 81.4±1.1 | 70.1±0.9 | 78.1±1.0 | 68.3±1.7 | - |
| Graph-Bert | 84.3±1.3 | 71.2±0.8 | 79.3±1.3 | - | - |
| SuperGAT$_{SD}$ | 82.7±0.6 | 72.5±0.8 | 81.3±0.5 | **74.4**±0.4 | 54.5±0.3 |
| DET | **84.6**±0.4 | 72.8±0.5 | **81.8**±0.3 | 74.1±0.3 | **55.7±0.3** |

Table 5: KG completion (entity prediction) results on FB15K-237 and WN18RR.

| Model | FB15K-237 | | | | WN18RR | | | |
|---|---|---|---|---|---|---|---|---|
| | MRR | MR | Hits@1 | Hits@10 | MRR | MR | Hits@1 | Hits@10 |
| TransE | .310 | 199 | .218 | .495 | .232 | 5,249 | .061 | .522 |
| RotatE | .338 | 177 | .241 | .533 | .476 | 3,340 | .428 | .571 |
| TuckER | .358 | - | .266 | .544 | .470 | - | .443 | .526 |
| CoKE | .364 | - | .272 | .549 | .484 | - | .450 | .553 |
| CompGCN | .355 | 197 | .264 | .535 | .479 | 3,533 | .443 | .546 |
| HittER | .373 | 158 | .279 | .558 | .503 | 2,268 | .462 | .584 |
| DET | **.376** | **150** | **.281** | **.560** | **.507** | **2,255** | **.465** | **.585** |

**Results**  The results are shown in Table 4, from which we can observe that DET outperformed the attention-based methods on most datasets except PPI. Although SuperGAT$_{SD}$ had better performance on this dataset, we argue that it is no contradiction to incorporate SuperGAT$_{SD}$ as structural encoder into DET to obtain a stronger model.

Interestingly, the attention-based methods unanimously performed worse than the GCN-based method GCN+NS on CiteSeer. SuperGAT$_{SD}$ and CGAT even had the same or worse results compared with the original GAT. Nevertheless, we observed an improvement from GAT to DET. This result empirically demonstrates the strength of leveraging semantic neighbors.

## 4.3 KG COMPLETION

**Datasets**  We conducted experiments on the KG completion (a.k.a., entity prediction) task. The main target is to predict the subject entity (or object entity) given an incomplete triple. We evaluated DET on two benchmark datasets FB15K-237 (Toutanova & Chen, 2015) and WN18RR (Dettmers et al., 2018), which are sampled from the real-world KGs Freebase (Bollacker et al., 2008) and WordNet (Miller, 1995), respectively.

**Baselines**  We chose the best-performing entity prediction methods as our baselines: the TransE-family methods TransE (Bordes et al., 2013), RotatE (Sun et al., 2019), and TuckER (Balazevic et al., 2019), and the attention-based methods CoKE (Wang et al., 2019b), CompGCN (Vashishth et al., 2020), and HittER (Chen et al., 2021b). Specifically, CoKE and HittER also leverage Transformer to encode the structural information.

**Results**  We report the main results on Table 5. It is clear that DET surpassed all the baselines across all datasets and metrics. The improvement on MR (mean rank) is most significant, which implies that DET learned better embeddings for all entities, not only for the top ones favored by Hits@1.

Overall, DET achieved competitive performance on all three types of tasks, which demonstrates its effectiveness and generality in modeling graphs.

Table 6: Ablation study results on different datasets (↑: higher is better; ↓: lower is better. ×: unavailable entry). St. and Se. are the abbreviations of Structural and Semantic.

| St. encoder | Se. encoder | Fetching loss | ZINC MAE↓ | Cora Accuracy↑ | CiteSeer Accuracy↑ | PubMed Accuracy↑ | FB15K-237 Mean Rank↓ | WN18RR Mean Rank↓ |
|---|---|---|---|---|---|---|---|---|
| ✓ | ✓ | ✓ | × | 84.6 | 72.8 | 81.8 | 150 | 2,255 |
| ✓ | ✓ | | 0.113 | 82.6 | 72.7 | 78.1 | 151 | 2,305 |
| ✓ | | | 0.122 | 84.0 | 71.6 | 80.9 | 158 | 2,268 |
| | ✓ | ✓ | × | 84.1 | 72.5 | 81.5 | × | × |
| | ✓ | | 0.515 | 83.1 | 72.6 | 77.7 | × | × |

Figure 3: Accuracy on three datasets with different homophily (Cora (Yang et al., 2016): 0.83, CoraFull (Bojchevski & Günnemann, 2018): 0.59, Chameleon (Rozemberczki et al., 2019): 0.21), w.r.t. hyper-parameter $\tau$ (average of 7 runs). The homophily statistics are from (Zhu et al., 2020).

## 5 FURTHER ANALYSIS

To better understand DET, we design three experiments to explore and evaluate DET in depth.

### 5.1 IS EVERY MODULE IN DET USEFUL?

We conducted ablation studies to verify the effectiveness of each module in DET. We used six datasets in different tasks and present the results in Table 6. We removed the modules from DET step-by-step while keeping identical hyper-parameter settings throughout the experiments.

**Semantic Neighbor Fetching** The semantic neighbor fetching loss is undoubtedly important to DET. No matter if combining two encoders or only using the semantic encoder, integrating with the semantic fetching module had better performance in most cases. The improvement was most notable on PubMed, where it yielded 3.7% and 3.8% of accuracy increases, respectively. The mean rank results on WN18RR also got worse without the fetching loss.

**Semantic Encoder** If we do not consider the semantic neighbor fetching loss, is the semantic encoder itself still useful to DET? Unfortunately, we find the answer ambiguous. For Cora, PubMed, and WN18RR, when we did not employ the fetching loss, DET with the semantic encoder performed much worse than DET without the semantic encoder. But we observe that the situation was reversed on CiteSeer and FB15K-237. In fact, when we only consider one-hop neighbors, the semantic encoder without fetching loss is just a "minus" attention layer. It may have its pros and cons compared with the standard dot-product attention layer on different datasets. In this sense, the semantic fetching loss is who endows the semantic encoder with the characteristic.

On the ZINC dataset, where the model can apply attention operations on the whole graph, the semantic encoder was capable of estimating the semantic similarity of remote neighbors to the node of interest without the help of the fetching loss. Therefore, we can see that the dual-encoding version of DET greatly outperformed the structural encoder only version. Overall, the effectiveness of the semantic encoder is conditioned: it must get in touch with the remote nodes.

**Structural Encoder** The structural encoder also has merits. From the results of the 3-rd and 5-th rows in Table 6, we find that it had better performance than the semantic encoder on all datasets except CiteSeer. We also noticed that only using the semantic encoder had the worst MAE on the ZINC dataset, due to the absence of all structural information.

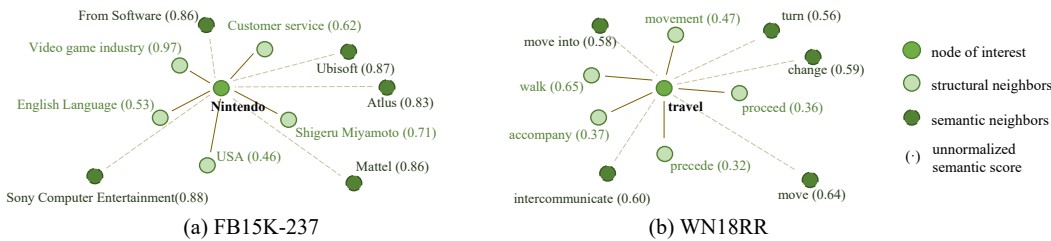

Figure 4: Examples of the semantic attention scores to different types of neighbors.

## 5.2 THE CORRELATION BETWEEN SEMANTIC ENCODING AND GRAPH HOMOPHILY

In the Introduction section, we mention that the semantic encoding is helpful even on graphs with high homophily. Therefore, we conducted experiment to verify the correlations between the effectiveness of the semantic encoder and the homophily of datasets.

We set a hyper-parameter $\tau$ to control the combination of the structural encoder and the semantic encoder, which can be written in the following equation:

$$\mathbf{h} = \tau \mathbf{h}^{\text{st}} + (1 - \tau)\mathbf{h}^{\text{se}}, \tag{8}$$

where $\mathbf{h}$, $\mathbf{h}^{\text{st}}$, $\mathbf{h}^{\text{se}}$ denote the combined output, structural output, and semantic output, respectively. By assigning different $\tau$, we can control the importance of each encoder in the combination.

The experimental results are shown in Figure 3, from which we can see that the performance gap for different $\tau$ existed in all three datasets, but the trends and peaks were different. On highly homophilic dataset Cora, the accuracy first increased from $\tau = 0$ to $\tau = 0.15$, and peaked around $\tau = 0.2$, then dropped until $\tau = 95$. On CoraFull and Chameleon with median and low homophily, we observe that the performance peaked around $[0, 0.05]$ and maintained steady in the interval $[0.05, 0.2]$. When $\tau \geq 0.5$, the performance rapidly dropped to minima. Therefore, we may make the following conclusions: (1) the semantic encoder has better effects when the datasets have low homophily. The larger proportion the semantic encoding occupied, the better performance the model achieved; (2) even on the highly homophilic dataset, engaging the semantic encoder with a proper $\tau$ has significant advantages over using only structural encoder; (3) in any cases, properly combining the output of two encoders (i.e., DET) is the best choice.

## 5.3 HOW DOES THE SEMANTIC ENCODER HELP THE STRUCTURAL ENCODER?

It is worth exploring how the semantic encoder affects the structural encoder. In Figure 4, we illustrate two examples on FB15K-237 and WN18RR, respectively. We find that the semantic scores for the structural neighbors are also in line with human intuition. In the left figure, the entity *USA* has a low score although it is directly connected to *Nintendo* by relation *service_location*. The verb *precede* and *accompany* obtain relatively low scores in the right figure. These neighbors are not very related to the entities of interest from the human perspective. By contrast, some one-hop neighbors get high semantic scores, e.g., the well-known director *Shigeru Miyamo* of *Nintendo* in FB15K-237 and the verb *walk* in WN18. They are the more informative entities. For the semantic neighbors, we can see that the exploited remote neighbors are closely related to the entity of interest, as well as the structural neighbors with high semantic scores. For example, *Atlus* is an important game developer to *Nintendo*. Aggregating such information may be helpful when the model is asked to predict the games related to *Nintendo*. For the verb *travel* in WN18RR, *move* also shares many key features with it.

## 6 CONCLUSION AND FUTURE WORK

In this paper, we propose a new Transformer architecture DET to deal with graphs of different types and sizes. In DET, the structural encoder aggregates information from local nodes while the semantic encoder seeks the remote nodes with useful semantics. The experimental results demonstrate the strong performance of DET on three prevalent GNN tasks across 9 benchmarks. We hope DET can bring more insights and inspirations in developing unified Transformer architectures. In future, we plan to adapt DET to the NLP and CV areas.

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

Table 7: The average training time of DET in comparison with respective baselines, on a 32GB V100.

| | PCQM4M-LSCv1 | ZINC | Cora | CiteSeer | FB15K-237 | WN18RR |
|---|---|---|---|---|---|---|
| Graphormer | 66h | 30h | - | - | - | - |
| SuperGAT | - | - | 4m | 6m | - | - |
| HittER | - | - | - | - | 27h | 14h |
| DET | 78h | 41h | 4m | 6m | 36h | 14h |

Table 8: The dataset statistics. $R$ and $C$ denote the regression and classification tasks, respectively.

| | PCQM4M-LSCv1 | ZINC | Cora | CiteSeer | PubMed | PPI | ogbn-arxiv | FB15K-237 | WN18RR |
|---|---|---|---|---|---|---|---|---|---|
| # Graphs | 3,803,453 | 12,000 | 1 | 1 | 1 | 24 | 1 | 1 | 1 |
| # Nodes | 53,814,542 | 277,920 | 2,708 | 3,327 | 19,717 | 56,944 | 169,343 | 14,951 | 40,943 |
| # Edges | 55,399,880 | 597,960 | 5,429 | 4,732 | 44,338 | 818,716 | 1,166,243 | 310,116 | 93,003 |
| # Edge-type | - | - | - | - | - | - | - | 237 | 11 |
| # Task | $R$ | $R$ | $C$ | $C$ | $C$ | $C$ | $C$ | $C$ | $C$ |
| # Classes | - | - | 7 | 6 | 3 | 121 | 40 | 14,951 | 40,943 |

## A  POSITION EMBEDDING

There are many important graph features that can be used to identify different nodes. Thanks to learnable position embedding, these discrete features now can be encoding into embeddings and combined with the raw nodes embedding vectors.

Specifically, for graph property prediction task, we use the method proposed by Graphormer (Ying et al., 2021) to encode degree centrality of an arbitrary node $v_i$ as follows:

$$\mathbf{c}_i = f_c(\deg(v_i)), \tag{9}$$

where $\deg(v_i)$ denotes the degree of the node $v_i$, and $f_c : \mathbb{R} \to \mathbb{R}^h$ is the mapping function that converts the node degree to a learnable embedding. We also consider encoding the distances from the node of interest $v_i$ to different neighbors by the following equation:

$$\mathbf{d}_{v_i,v_j} = f_d(\text{spd}(v_i, v_j)), \tag{10}$$

where $\text{spd}(v_i, v_j)$ denotes the shortest path distance from $v_i$ to $v_j$, and $f_d : \mathbb{R} \to \mathbb{R}^h$ is a similar function that converts the distance to a learnable embedding.

For KG representation learning task, we follow (Chen et al., 2021b) to encode the edge types into node embeddings, which is implemented by an additional atom triple Transformer $\mathcal{M}_A$. Specifically, for a given triple $(v_i, r_{ij}, v_j)$ for the node of interest $v_i$, where $r_{ij}$ denote the edge type (i.e., relationship) between $v_i$ and $v_j$. The edge type information can be encoded by the following equation:

$$\mathbf{e}_{ij} = \mathcal{M}_A([\mathbf{c}_A, \mathbf{v}_i, \mathbf{r}_{ij}, \mathbf{v}_j]), \tag{11}$$

where $[\mathbf{c}_A, \mathbf{v}_i, \mathbf{r}_{ij}, \mathbf{v}_j]$ is the input embedding sequence. $\mathbf{c}_A$ is the virtual node for the atom Transformer, whose output represents the edge encoding embedding.

## B  COMPUTATIONAL COST

We used one 32GB V100 GPU to train all the methods for estimation and show the average training time in Table 7. Clearly, incorporating the semantic encoder did not significantly increase the computational cost, especially on the node classification datasets.

## C  EXPERIMENT DETAILS

### C.1  DATASET SETTINGS

We present the overall dataset statistics in Table 8.

Table 9: The hyper-parameter settings on the datasets in the main experiments.

| | PCQM4M-LSCv1 | ZINC | Cora | CiteSeer | PubMed | PPI | ogbn-arxiv | FB15K-237 | WN18RR |
|---|---|---|---|---|---|---|---|---|---|
| Learning-rate | 0.0002 | 0.0002 | 0.005 | 0.005 | 0.01 | 0.005 | 0.05 | 0.01 | 0.008 |
| Batch-size | 256 | 256 | 128 | 128 | 128 | 128 | 128 | 512 | 512 |
| # Layer | 12 | 12 | 2 | 2 | 2 | 2 | 2 | 6 | 6 |
| # Multi-heads | 32 | 8 | 8 | 8 | 8 | 8 | 8 | 8 | 8 |
| $\tau$ | increasing | increasing | 0.15 | 0.15 | 0.15 | 0.15 | 0.15 | 0.5 | 0.5 |
| # Semantic neighbors | - | - | 16 | 16 | 16 | 16 | 16 | 50 | 50 |
| # Negative samples | - | - | 64 | 64 | 64 | 64 | 64 | 50 | 50 |

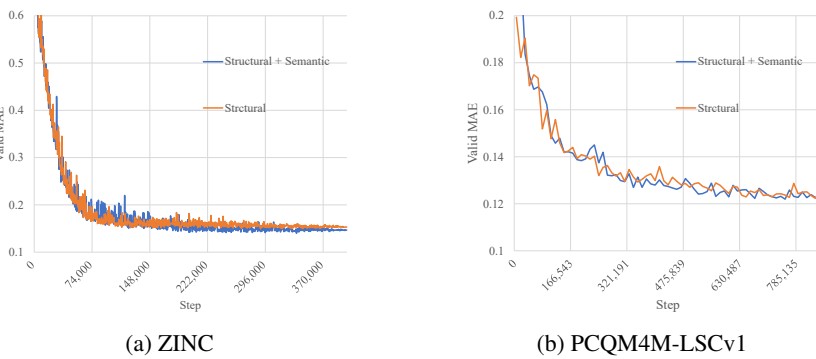

(a) ZINC  (b) PCQM4M-LSCv1

Figure 5: The validate MAE on ZINC and PCQM4M-LSCv1, w.r.t., training step.

**Graph Property Prediction** For PCQM4M-LSCv1, the model is asked to predict the DFT (density functional theory)-calculated HOMO-LUMO energy gap of given molecules. It contains more than 3.8M 2D molecular graphs as input, which is especially appropriate to evaluate the performance of model in large scale scenarios. On other hand, ZINC is a relative small datasets, where the main target is to predict the graph property regression for constrained solubility. It is one of the most popular real-world molecular datasets for graph representation learning.

**Node Classfication** Cora, CiteSeer and PubMed are three citation network datasets proposed by (Yang et al., 2016). They are typically used for transductive node classification task. PPI on the other hand is used for inductive evaluation. It consists of 24 graphs, with 20 graphs for training, and 2 for validation and 2 for testing. All four datasets are the prevalent benchmarks used for node classification.

**KG Completion** FB15K-237 is the revised version of the original FB15K dataset (Bordes et al., 2013) that was used as entity prediction benchmark in last ten years. However, recent studies (Dettmers et al., 2018; Toutanova & Chen, 2015) find that the original FB15K contains a large proportion of redundant data, some of which may incur testing data leakage, the same to another well-used dataset WN18. Therefore, most latest studies only use the revised datasets FB15K-237 and WN18RR for evaluation. FB15K-237 has more different relationships, while WN18RR is more sparse and has more different entities.

## C.2 PARAMETER SETTINGS

We summarize the main hyper-parameter settings on different datasets in Table 9. The more specific settings can be found in the source code. For the graph property prediction tasks, we directly let the semantic encoder access all neighbors, and thus the semantic neighbor fetching loss is not used. Alternatively, we adapt an increasing strategy to gradually improve the weight of the output of semantic encoder during training.

Table 10: Accuracy on 8 popular node classification datasets.

| Model | CS | Physics | Cora-ML | Cora-Full | DBLP | Chameleon | Four-Univ | Wiki-CS |
|-------|-----|---------|---------|-----------|------|-----------|-----------|---------|
| GCN | $\mathbf{91.5}_{\pm\mathbf{0.2}}$ | $\mathbf{92.5}_{\pm\mathbf{0.2}}$ | $\mathbf{85.0}_{\pm\mathbf{0.4}}$ | $59.5_{\pm0.2}$ | $77.8_{\pm0.5}$ | $33.1_{\pm0.9}$ | $74.8_{\pm0.6}$ | $74.0_{\pm1.0}$ |
| GraphSAGE | $90.0_{\pm0.1}$ | $92.2_{\pm0.1}$ | $83.7_{\pm0.4}$ | $59.2_{\pm0.2}$ | $78.7_{\pm0.6}$ | $41.0_{\pm0.9}$ | $74.4_{\pm0.6}$ | $77.5_{\pm0.5}$ |
| GAT | $89.5_{\pm0.2}$ | $91.2_{\pm0.6}$ | $83.2_{\pm0.6}$ | $58.7_{\pm0.3}$ | $78.2_{\pm1.5}$ | $40.8_{\pm0.7}$ | $74.2_{\pm0.7}$ | $77.6_{\pm0.6}$ |
| SuperGAT$_{\text{SD}}$ | $88.8_{\pm0.4}$ | $91.6_{\pm0.5}$ | $84.5_{\pm0.4}$ | $55.8_{\pm0.6}$ | $79.4_{\pm0.8}$ | $41.6_{\pm0.7}$ | $76.2_{\pm0.8}$ | $77.9_{\pm0.7}$ |
| DET | $89.9_{\pm0.1}$ | $92.0_{\pm0.4}$ | $84.5_{\pm0.4}$ | $\mathbf{59.9}_{\pm\mathbf{0.2}}$ | $\mathbf{80.1}_{\pm\mathbf{0.4}}$ | $\mathbf{41.8}_{\pm\mathbf{0.7}}$ | $\mathbf{76.4}_{\pm\mathbf{1.0}}$ | $\mathbf{78.5}_{\pm\mathbf{0.4}}$ |

Table 11: A result comparison of different $f_s$.

| Methods | Cora | CiteSeer | PubMed |
|---------|------|----------|--------|
| The proposed | 84.6 | 72.8 | 81.8 |
| Linear | 83.3 | 72.3 | 80.2 |

# D ADDITIONAL EXPERIMENTS

## D.1 ANALYSIS ON THE TRAINING PROCESS

We conducted experiments to compare the performance of structural encoder with and without the semantic encoder, in term of training steps.

We depict the valid MAE results on ZINC in Figure 5a. At the beginning of the training, we find that the two methods do not have a visible performance gap. The curves are tightly overlapped during step 0 to 74, 000.

As the performance starts to be converged, i.e., step 74, 000 to 148, 000, only using structural encoder is better than combining the two encoders. However, as the valid MAE tends to be stable, the dual-encoding DET gradually outperforms the single structural encoder.

It is worth-noting that the turning point appears at the performance starts to be converged, where the input embeddings also approach the ideal positions. Therefore, the semantic similarity among embeddings can be estimated more precisely, and contribute to a better semantic encoder. Therefore, DET can obtain a lower MAE than the single strctural encoder. We present the result on PCQM4M-LSCv1 in Figure 5b, which supports the same conclusion.

## D.2 RESULTS ON OTHER NODE CLASSIFICATION DATASETS

We also provide the results on additional 8 real-world datasets for node classification. They are: CS and Physics (Shchur et al., 2018); Cora-ML, Cora-Full and DBLP (Bojchevski & Günnemann, 2018); Chameleon (Rozemberczki et al., 2019); Four-Univ (Craven et al., 1998); and Wiki-CS (Mernyei & Cangea, 2020). We report the accuracy on Table 10. It is clear that DET consistently and significantly outperformed the attention-based methods on these datasets.

## D.3 CHOICES OF $f_s$

In this section, we conducted experiments to verify the effectiveness of proposed $f_s$, in comparison with a linear attention implementation. The results are shown in Table 11.we can observe that our proposed attention outperformed linear attention, and the results of using linear attention were similar to those of SuperGAT and the original GAT. SuperGAT also leverages a self-supervised contrastive loss to predict the existence of edges in the original graph. If we use linear attention to replace our proposed attention, our method is more like a SuperGAT with two separated linear attentions to cope with node classification and edge prediction, respectively.

