# OpenReview forum: "Reach the Remote Neighbors: Dual-Encoding Transformer for Graphs"
_ICLR.cc/2023/Conference — Submitted to ICLR 2023_

### Official Review · Reviewer_FttT · 2022-10-21

**Confidence:** 3
**Correctness:** 3
**Technical Novelty And Significance:** 3
**Empirical Novelty And Significance:** 3
**Recommendation:** 5

**Clarity, Quality, Novelty And Reproducibility:**

Clarity: Generally, this paper is well-written. One suggestion is to notate the structural and semantic embeddings (i.e., $h$) with layer index (e.g., by superscript) and explain how the transformer layers are stacked.

Quality: The proposed novel architecture is motivated in a natural way. The design of the semantic component sounds good from a technical perspective, but a related theoretical analysis would be more in a demand. The proposed architecture is evaluated on various levels of tasks, and its advantages over SOTA methods seem to be consistent. It would be better to provide standard deviations and a statement of whether the advantage is statistically significant.

Novelty: The proposed neural architecture is novel to me, although it is somewhat straightforward.

Reproducibility: All the adopted datasets, considered baselines, and the implementations are publicly available.

**Strength And Weaknesses:**

strengths:
1. This paper studies a critical problem, that is to say, how to preserve the advantage of transformer to enjoy a large receptive field and, at the same time, efficiently aggregate discriminative information. In this regard, the contribution is likely to make a significant impact to the community.
2. The proposed neural architecture is novel to me, which, instead of considering sparsity trick or hierarchical structure, directly defines neighborhood in the latent semantic space.
3. The experiments are solid in terms of their comprehensiveness. Besides, the ablation studies confirm the effectiveness of introduced components. The case study of semantic neighbors also implies that the semantic encoder works with meaningful neighbors.

weaknesses:
1. It seems that there is a lack of theoretical analysis about the proposed architecture. I am really curious about what semantic similarities would be learned eventually. Meanwhile, what is the negative distribution should be adopted deserves a more detailed discussion, especially considering the number of examples increases along with the number of hops exponentially.

**Summary Of The Paper:**

This paper proposes a novel graph transformer architecture that jointly aggregates structural neighbors and semantically related neighbors to encode each node. The proposed method is evaluated on various levels of graph learning tasks and confirmed to be effective.

**Summary Of The Review:**

I think the proposed transformer architecture is somewhat novel to me. It is empirically evaluated in an extensive way, showing its effectiveness. Currently, I can only evaluate the behavior of semantic component based on the case shown in experiment, where the semantic neighbors seem to be truly semantically relevant. I don't have confidence to suggest an acceptance now, and more theoretical analysis would resolve my concerns.

---

> ### Author Response · Authors · 2022-11-09
> **Response to Reviewer FttT**
>
> Many thanks for the detailed and constructive feedback. We hope the following replies can erase your concerns:
>
> #### **Weakness**
>
> 1. **It seems that there is a lack of theoretical analysis about the proposed architecture. I am really curious about what semantic similarities would be learned eventually. Meanwhile, what is the negative distribution should be adopted deserves a more detailed discussion, especially considering the number of examples increases along with the number of hops exponentially.**
>
>     We are sorry for the absence of theoretical analysis. Honestly, our idea is inspired by the recent MSA Transformer that makes use of family member information and row and column attentions. As it is very straightforward, we exhausted its empirical performance across multiple types of graphs. Therefore, we argue that the lack of theoretical analysis won’t hurt its effectiveness.
>
>     We thank you very much for the suggestion towards negative sampling that is also pointed out by Reviewer aSXN. We have updated an experiment to compare different score functions (i.e, $f_s$). Although our current implementation uses only one-hop neighbors as local neighbors, we believe that it is not a non-trivial problem for multi-hop cases. We can follow existing methods to adopt some sampling strategies and only consider a small number of multi-hop neighbors.

---

> > ### Comment · Reviewer_FttT · 2022-11-26
> > **Discussion**
> >
> > Thanks for your response! I think this paper is ok but has not been above the bar of ICLR. I encourage the authors to make the proposed method more principled.

---

> > > ### Author Response · Authors · 2022-11-28
> > > **Response to Reviewer FttT**
> > >
> > > Hi, thank you very much for your kind reply. Our method may lack some theoretical analysis but it is a principled method (as it can be generalized to different Transformer variants and applied to different graphs and tasks).

---

### Official Review · Reviewer_e1hj · 2022-10-21

**Confidence:** 4
**Correctness:** 3
**Technical Novelty And Significance:** 2
**Empirical Novelty And Significance:** 2
**Recommendation:** 5

**Clarity, Quality, Novelty And Reproducibility:**

The clarity of the paper is not satisfactory, certain modules are not very clear to me.
The authors provide source code that looks well documented so I guess there's no concern on reproducibility.
I feel there's quite limited novelty in the proposed approach. The dual module idea is similar to model ensemble.

**Strength And Weaknesses:**

strength:
- The idea of disentangle and combine structural neighbors and semantic neighbors looks interesting to me.
- The authors conducted thorough experiments on various datasets.

weaknesses:
- Some writing is a bit confusing. How to understand and interpret Figure 2? How do the authors define the semantic set $\mathcal{N}^{\text{se}}$? "In our implementation, it is sampled from the top candidate" what are the pool to select top candidates then? Is it all the other nodes?
- What's the runtime analysis of the proposed method? How to understand Table 7? Why sometimes there's a nontrivial overhead and sometimes not?
- The empirical performance of the proposed method is no longer the state-of-the-art performance. Please check [1]'s results on ZINC, for example. The authors should include and compare with this paper.

[1] Dwivedi, Vijay Prakash, et al. "Graph Neural Networks with Learnable Structural and Positional Representations." International Conference on Learning Representations. 2022.

**Summary Of The Paper:**

This paper proposes a new Graph Transformer architecture, named dual-encoding Transformer (DET), which has a structural encoder to aggregate information from near neighbors and a semantic encoder to focus on useful semantically close neighbors. The empirical results demonstrate that the proposed DET achieves superior performance compared to the respective state-of-the-art attention-based methods.

**Summary Of The Review:**

The submission has thorough empirical results, but the empirical results do not look very competitive and the proposed method has only limited novelty

---

> ### Author Response · Authors · 2022-11-09
> **Response to Reviewer e1hj**
>
> #### **Weakness**
>
> - **Some writing is a bit confusing. How to understand and interpret Figure 2? How do the authors define the semantic set? "In our implementation, it is sampled from the top candidate" what are the pool to select top candidates then? Is it all the other nodes?**
>
>     Sorry for the confusion. In figure 2, our basic idea is to view node embeddings as protein amino acid sequences with a fixed length. Then, we can make use of the insight from the well-known MSA Transformer and AlphaFold 2. To this end, we need to find the family members (that share similar evolutional characteristics and relationships) of our “protein’’. In their original implementation, this step is done by querying an external gene database. In our case, we propose the semantic encoder to find such family members. We have updated these explanations in the revision.
>
>     We select the top 10% of nodes as the candidate pool for a general setting (see Section 3.4) and then sample the training semantic neighbors from it. The number of semantic neighbors can be found in Table 9 of the Appendix.
>
> - **What's the runtime analysis of the proposed method? How to understand Table 7? Why sometimes there's a nontrivial overhead and sometimes not?**
>
>     The runtime may be correlated with the scale of datasets. For example, in KG completion, the size of FB15K-237 is much bigger than WN18RR. Optimizing DET may cost more time than the method leveraging only structural information.
>
> - **The empirical performance of the proposed method is no longer the state-of-the-art performance. Please check [1]'s results on ZINC, for example. The authors should include and compare with this paper.**
>
>     Thanks very much for the reference. We have updated the corresponding table. As we used Graphormer as the structural encoder, the comparison is still fair and significant to demonstrate the effectiveness of dual encoding. It is also possible to integrate DET with the learnable position embedding proposed in [1].
>
> #### **Summary Of The Review**
>
> - **The submission has thorough empirical results, but the empirical results do not look very competitive and the proposed method has only limited novelty**
>
>     We also noticed this phenomenon in the node classification tasks, which is why we provide the mean and std results. Overall, our method has good generalization and is competitive for dealing with multiple types of graphs. Our code also uses the SOTA implementation as the structural encoder (i.e., Graphormer, GAT, and HittER). See the details in the uploaded code and Appendix A. Therefore, we ensured a fair comparison between DET and all baselines.

---

### Official Review · Reviewer_aSXN · 2022-10-23

**Confidence:** 3
**Correctness:** 3
**Technical Novelty And Significance:** 3
**Empirical Novelty And Significance:** 3
**Recommendation:** 6

**Clarity, Quality, Novelty And Reproducibility:**

Clarity:
The paper is generally well-written with a few caveats:
- The motivation for the SSL loss is not clear: why enforcing high similarity between neighboring nodes? At first sight it seems redundant with the structural attention scheme which only applies attention to neighbors.
- If I understand correctly, for a given graph, DET has to sort $n$ times $n$ scores and store $n^2$ scores in memory?

Quality:
- The methods is clearly described, correctly motivated and simple.
- Experimental protocol seems thorough: varied tasks (graph and node level) as well as baselines are used. Moreover, sensible ablations are conducted although one seems to be missing in my opinion (see weakness).

Novelty:
- DET relies on two attention mechanisms: a local one, aimed at reflecting the structure, and a sparse, global one, aimed at gathering semantically similar remote nodes. Both mechanisms seem at least partially novel to me. Encoding the structure by considering substructures such as neighboring nodes has already been proposed [1][2] without involving a virtual node, and could potentially be discussed to better delineate the contribution of this work.

[1] GraphiT: Encoding Graph Structure in Transformers (Mialon et al. (2021))

[2] Structure-Aware Transformer for Graph Representation Learning (Chen et al. (2022))

Reproducibility: the code is provided along with experimental details in the appendix.


**Strength And Weaknesses:**

Strength:
- This work offers a potential solution, DET, to an important problem, scaling transformers to large graphs.
- The method is simple and motivated: a local attention scheme is augmented by an attention scheme that is able to fetch remote useful nodes. An SSL loss allows to learn the score used for ranking and attention weighting. Ranking only every few epochs seems to provide an efficient attention scheme in practice.
- DET demonstrates strong results on various and important benchmark such as PMQC4M and ZINC.
- This work studies in detail the proposed attention scheme and in particular the semantic attention: we see that the nodes fetched by the semantic encoder make sense, and that the semantic encoder is as expected useful when considering low homophily graphs.

Weaknesses:
- One novelty of this work, and what makes it scalable, is applying a custom attention to top ranked nodes, with an updated ranking every few epochs. In light of this, a natural baseline would be to apply linear attention instead of the proposed $f_s$: it would be more convincing to demonstrate the usefulness of $f_s$ compared to linear attention.
- The other novelty is the use of a virtual node for a local attention mechanism. I am not sure to understand the motivation of this mechanism compared to a purely local attention scheme (attention between the node of interest and its neighbors): clarification and/or ablation could be useful to better understand the interest of the proposed scheme.

**Summary Of The Paper:**

This work proposes DET, a transformer architecture for handling large graphs.

DET relies on two attention mechanisms:
- A structural attention mechanisms, i.e. for a given node, attention applied between a virtual global node and the neighboring nodes only. Moreover, a learnable position encoding is added (including centrality, distance to other neighbors and edge type).
- A semantic attention mechanism, i.e. attention applied to semantic neighbors. More precisely, and for a given node, the other nodes are ranked depending on a learned embedding similarity score which is then used as an attention weight. This score can be learned via a self-supervised objective.
Although computing the similarity score for each node has quadratic complexity in the number of nodes., the ranking step is done every few epochs only, providing an efficient attention mechanism in practice.

The authors then proceed to demonstrate the effectiveness of graph and node level tasks, with strong results for PCQM4M, ZINC, ogbn arxiv and knowledge graph completion. Finally, further studies of DET are provided: an ablation is done to understand the role of structural attention, semantic attention, and the semantic attention loss, a study on the usefulness of semantic encoding w.r.t. graph homophily, and an example of learned relations by the different mechanisms.


**Summary Of The Review:**

This work offers a sensible solution to an important problem, with strong results. Although some elements could be clarified in the writing (novelty, more motivation for the virtual node in local attention and the SSL loss) and some ablation may be in my opinion missing, the pros outweight the cons and I tend to recommend acceptance.

---

> ### Author Response · Authors · 2022-11-09
> **Response to Reviewer aSXN**
>
> Thank you very much for the valuable comments. We have addressed the issues in the revision. Please find our answers to your questions below:
>
>
> #### **Weakness**
>
> -  **One novelty of this work, and what makes it scalable, is applying a custom attention to top ranked nodes, with an updated ranking every few epochs. In light of this, a natural baseline would be to apply linear attention instead of the proposed: it would be more convincing to demonstrate the usefulness of compared to linear attention.**
>
>     Thanks very much for your suggestion. We will update the detailed results later because training models for graph property prediction and KG completion tasks would cost several days. Here, we provide some results on node classification tasks to compare the proposed attention with the linear attention.
>
>     From the following table, we can observe that our proposed attention outperforms linear attention, and the results of using linear attention are similar to those of SuperGAT and the original GAT. SuperGAT also leverages a self-supervised contrastive loss to predict the existence of edges in the original graph. If we use linear attention to replace our proposed attention, our method is more like a SuperGAT with two separated linear attentions to cope with node classification and edge prediction, respectively.
>
> Methods | Cora | CiteSeer | PubMed |
> ----|----|----|----|
> The proposed | 84.6 | 72.8 | 81.8 |
> Linear | 83.3 | 72.3 | 80.2 |
> Original GAT | 83.0 | 72.5 | 79.0 |
> SuperGAT | 82.7 | 72.5 | 81.3 |
>
> - **The other novelty is the use of a virtual node for a local attention mechanism. I am not sure to understand the motivation of this mechanism compared to a purely local attention scheme (attention between the node of interest and its neighbors): clarification and/or ablation could be useful to better understand the interest of the proposed scheme.**
>
>     Sorry for the misunderstanding, and we will clarify this. Leveraging a virtual node to aggregate local neighbors is not our contribution. Instead, it is widely used in many existing papers towards graph representation learning with Transformer. We hereby highlight our contribution and motivation again: we propose to encode two types of neighbors for Transformer-based graph representation learning, which enables the standard Transformer to attend the nodes at arbitrary distance. The process of fetching remote neighbors is similar to querying the gene database in MSA Transformer. The dual encoding process is also inspired by the row and column attention mechanisms in MSA Transformer and AlphaFold2.
>
>
> #### **Clarity**
> - **The motivation for the SSL loss is not clear: why enforcing high similarity between neighboring nodes? At first sight it seems redundant with the structural attention scheme which only applies attention to neighbors.**
>
>     Sorry for the misunderstanding, and we will clarify this. The main goal of the SSL loss is to learn a desired $f_s$, rather than force high similarity between the input two node embeddings. We use local neighbors as positive examples, in which a distant node that possesses similar characteristics with local neighbors will also be “positive’’. Furthermore, $f_s$ also considers the similarity between two input embeddings as we believe the nodes with similar embeddings are helpful to encode the node of interest, inspired by AlphaFold and MSA Transformer.
>
> - **If I understand correctly, for a given graph, DET has to sort times scores and store scores in memory?**
>
>     We did not store the full score matrix during training. Instead, we only keep a small number of candidate semantic neighbors in memory for efficiency (please see Table 9 for parameter settings).

---

> > ### Comment · Reviewer_aSXN · 2022-12-12
> > **Rebuttal seen**
> >
> > Thank you for your rebuttal. I appreciate the clarification on $f_s$ and linear attention in Appendix D.3. Sorry for the misunderstanding on the virtual node. If I understand correctly the key contribution is to separate a local self-attention scheme (structural) from a global sparse one (global). The second mechanism is attention applied to the top candidate according to a learned score function $f_s$. If so, I think the novelty to be slightly limited and keep my score unchanged, I still think the paper is above average

---

### Official Review · Reviewer_4oiv · 2022-10-25

**Confidence:** 5
**Correctness:** 2
**Technical Novelty And Significance:** 2
**Empirical Novelty And Significance:** 2
**Recommendation:** 3

**Clarity, Quality, Novelty And Reproducibility:**

**************Clarity**************

The paper is clearly written to understand overall method.

**************Quality**************

Some experiments for baselines are not properly conducted. Also, it would be better to conduct more experiments to validate the effectiveness of proposed models.

**********Novelty**********

The proposed method has a limited novelty.

******************************Reproducibility******************************

The authors share their source code in the supplement. The paper has good reproducibility.

**Strength And Weaknesses:**

**Strengths**

(1) This paper addresses the limitation of Transformer-based models for graphs, which is one of important topics to generalize Transformer to the graph domain.

(2) The proposed DET shows the best performance on KG completion task.

**Weakness**

(1) I think that the novelty of this paper is limited.

- The combination of structurally and semantically close neighbors has been already discussed in various works (Geom-GCN [1], Non-local GNNs [2]).
- Also, semantic neighbor fetching loss seems similar to self-supervised loss of SuperGAT [3].

(2) The accuracy of GCN and GraphSAGE on ogbn-arxiv is very low. From the leaderboard of ogbn-arxiv, the accuracy of GCN and GraphSAGE are 71.74 and 71.49, respectively. What makes deteriorating performance?

(3) From Table 7, DET requires more average training time compared to Graphormer on PCQM4M-LSCv1 and ZINC datasets although DET uses less neighborhoods compared to Graphormer.

(4) In Section 5.2, the paper discussed the correlation between semantic encoding and graph homophily. So, it would be better if the paper compare the performance on heterophilic graph datasets with baselines such as H2GCN [4].

---

[1] Pei, Hongbin, et al. "Geom-gcn: Geometric graph convolutional networks." ICLR 2020.

[2] Liu, Meng, Zhengyang Wang, and Shuiwang Ji. "Non-local graph neural networks." TPAMI 2021.

[3] Kim, Dongkwan, and Alice Oh. "How to find your friendly neighborhood: Graph attention design with self-supervision." ICLR 2021.

[4] Zhu, Jiong, et al. "Beyond homophily in graph neural networks: Current limitations and effective designs." NeurIPS 2020.

**Summary Of The Paper:**

The paper proposes Dual-Encoding Transformer (DET), which aggregates information from both local neighbors and semantically close neighbors to address the scalability issue of existing Transformer architectures for graphs. Also, the authors design self-supervised semantic neighbor fetching loss, which is jointly optimized with the main task loss.

**Summary Of The Review:**

Overall, I am leaning towards rejection. My major concern is the novelty and experimental results. If you address my concerns, I will raise my score.

---

> ### Author Response · Authors · 2022-11-09
> **Response to Reviewer 4oiv**
>
> Thank you very much for the detailed and constructive comments. We have revised the paper accordingly. Please find our answers to your questions below:
>
> ## Weakness
>
> #### **(1) The novelty is limited.**
>
> - **The combination of structurally and semantically close neighbors has been already discussed in various works (Geom-GCN [1], Non-local GNNs [2]).**
>
>     Many thanks for your suggested references. Geom-GCN and Non-local-GNNs are indeed relevant to our paper, especially Non-local GNNs that learns a sort of non-local node embeddings (may also be useful for our method to reduce computational cost). We have added them to the new revision (marked in red in the related work section). We are sorry for the missing references, but we still believe our method DET has its own merits.
>
>     Specifically, Geom-GCN learns the aggregation purely based on embedding distance, while Non-local-GNNs uses the attention scores from a virtual node to other nodes as a sorting metric to find non-local neighbors. In this sense, our method DET exploits the complementary strengths of two methods.
>
>     But, in fact, our motivation is different from theirs. The definition of distant and informative nodes, to Geom-GCN,  is the nodes with similar embeddings; to Non-local-GCNs, is the nodes with similar attention scores; and to our DET, is the remote nodes that function like one-hop neighbors and are close to the input node in embedding space.
>
>     Moreover, DET is tailored to Transformer and generalized to many different types of graphs. Its learning tasks are across regression, multi-classification, and multi-classification with a large number of classes (i.e., KG completion). By contrast, the above two related works only focus on modeling networks and are evaluated on multi-classification tasks with a few classes (also see Table 8 for dataset details).
>
>     Last but not least, the implementation is distinct. The two related works do not leverage a dual attention mechanism. They do not distinguish between remote nodes and direct neighbors.
>
> - **Also, semantic neighbor fetching loss seems similar to self-supervised loss of SuperGAT [3].**
>
>     Our loss is different from that of SuperGAT. Although both losses are based on contrastive learning, their sample sets and objectives are different. The loss in SuperGAT is used to predict the existence of an edge between two nodes, which is more like a multi-task learning setting. By contrast, the fetching loss in DET is used to find remote neighbors. Based on the definition of $f_s$, it exploits not only the underlying edges but also the close nodes in embedding space.
>
> #### **(2) The accuracy of GCN and GraphSAGE on ogbn-arxiv is very low. What makes deteriorating performance?**
>
> Sorry for the misunderstanding. The original results on obgn-arxiv were produced using SuperGAT. We also notice that the accuracy of GAT in this implementation is lower than that on the leaderboard. Considering our goal is to design a Transformer architecture for diverse graphs, we did not code from scratch. Following SuperGAT, we use a popular GitHub repository pyGAT to develop our method to ensure a fair comparison (at least among GAT, SuperGAT and ours). Although our experimental observations may still hold, we are transferring our code to a better GNN framework (the results may be released a few days later). Thank you very much for your helpful suggestion.
>
>
> #### **(3) From Table 7, DET requires more average training time compared to Graphormer on PCQM4M-LSCv1 and ZINC datasets although DET uses less neighborhoods compared to Graphormer.**
>
> For the graph property prediction task, where a graph usually has less than 200 nodes, both DET and Graphormer used all nodes as neighbors (as explained in Section 4.1). The semantic encoder module in DET caused the extra training time.
>
> #### **(4) In Section 5.2, the paper discussed the correlation between semantic encoding and graph homophily. So, it would be better if the paper compare the performance on heterophilic graph datasets with baselines such as H2GCN [4].**
>
> Thanks for your suggestion. The datasets used in Section 5.2 are selected based on their homophily statistics in H2GCN [4]. Besides, knowledge graphs (KGs) can be viewed as ``super heterophilic graphs’’, and we also conducted experiments on two KG completion benchmarks.

---

> > ### Comment · Reviewer_4oiv · 2022-11-27
> > **Thank you for your response.**
> >
> >
> > Thank you for your response on my review. I still have concerns as below:
> >
> > >But, in fact, our motivation is different from theirs. The definition of distant and informative nodes, to Geom-GCN, is the nodes with similar embeddings; to Non-local-GCNs, is the nodes with similar attention scores; and to our DET, is the remote nodes that function like one-hop neighbors and are close to the input node in embedding space.
> >
> > I would like to highlight that graph-based methods using semantically close neighbors have been already discussed in various works. As you mentioned, I already know that Geom-GCN defines neighbors based on embedding distance and Non-local GNN uses the attention scores for defining neighbors. But, the point is that they use semantically close neighbors as well as structurally close neighbors.
> > In addition, papers for graph structure learning such as IDGL [1]  define semantically close neighbors as nodes that are close to the input node in embedding space. So, in this reason, it is hard to say the paper proposed a novel framework for me.
> >
> > >Besides, knowledge graphs (KGs) can be viewed as ``super heterophilic graphs’’, and we also conducted experiments on two KG completion benchmarks.
> >
> > Knowledge graphs can be viewed as “heterogenous graphs”, which are different from “heterophilious graphs”.
> > As noted on H2GCN [2], the graph is heterogeneous if it has at least two types of nodes and different relationships between them, and homogeneous if it has a single type of nodes and a single type of edges. The type of nodes in heterogeneous graphs doe not necessarily match the class labels, therefore both homogeneous and heterogeneous graphs may have different levels of homophily. So, “heterophily” is a distinct concept from “heterogeneity”.
> >
> >
> > ---
> > [1], Chen, Yu, Lingfei Wu, and Mohammed Zaki. "Iterative deep graph learning for graph neural networks: Better and robust node embeddings." Advances in neural information processing systems 33 (2020): 19314-19326.
> > [2], Zhu, Jiong, et al. "Beyond homophily in graph neural networks: Current limitations and effective designs." Advances in Neural Information Processing Systems 33 (2020): 7793-7804.

---

> > > ### Author Response · Authors · 2022-11-28
> > > **Thank you very much for your detailed response.**
> > >
> > > It may be unfair to underestimate the novelty of our paper according to the existence of several related works. As aforementioned, our method is novel in both motivation and implementation, and capable of applying to diverse graph representation learning tasks. The reviewer may overfocus on one type of graph and overlooked the pros of this paper towards different types of graphs and tasks.
> > >
> > > For the second problem, we argue that knowledge graphs are still ``super heterophilic graphs’’ under that definition. The connected two entities in a KG do not belong to the same class almost everywhere.

---

### Decision · Program_Chairs · 2023-01-20

**Decision:**

Reject

**Justification For Why Not Higher Score:**

Would like to see most reviewer concerns around experiments resolved.

**Justification For Why Not Lower Score:**

N/A

**Metareview: Summary, Strengths And Weaknesses:**

This paper proposes a modified Transformer architecture for graphs that leverages neighbors close in graph space and in representation space. While reviewers are in agreement that this is an important direction, the general view is that the novelty relative to other related works in this direction is not particularly high, so there is more of a burden on the experimental results, and there remain a number of concerns, including additional comparisons to baselines and anomalies in the reported results.